# Socially Sustainable Interventions for Childhood Obesity Management: A Scoping Review of Randomized Controlled Trials

**DOI:** 10.3390/healthcare13222932

**Published:** 2025-11-16

**Authors:** Regiane de Paula, Vitor de Salles Painelli, Leonardo Vidal Andreato, Braulio Henrique Magnani Branco, Rúbia Gomes Corrêa

**Affiliations:** 1Postgraduate Program in Clean Technologies, Cesumar University (UniCesumar), Maringa 87050-390, PR, Brazil; regipaula74@gmail.com (R.d.P.); rubia.correa@unicesumar.edu.br (R.G.C.); 2Postgraduate Program in Health Promotion, Cesumar University (UniCesumar), Maringá 87050-390, PR, Brazil; vidal.leo@hotmail.com (L.V.A.); braulio.branco@unicesumar.edu.br (B.H.M.B.); 3Postgraduate Program in Movement Science, State University of Piauí, Teresina 64049-550, PI, Brazil; 4Strength Training Study and Research Group, Institute of Health Sciences, Paulista University, São Paulo 01533-000, SP, Brazil; 5Cesumar Institute of Science, Technology and Innovation—ICETI, Maringa 87050-390, PR, Brazil

**Keywords:** pediatric obesity, social sustainability, body mass index

## Abstract

Background/Objectives: Childhood obesity is a pressing global health issue. Addressing this multifaceted issue requires comprehensive interventions, particularly those improving social sustainability by strengthening support systems in families, schools, and communities. This scoping review explored interventions aimed at enhancing social sustainability to improve anthropometric outcomes in overweight and obese children. Methods: A literature search was conducted from 2 August to 1 September 2025 and included all studies published up to the latter date using PubMed, Scopus, Virtual Health Library, and Cochrane Library databases. Inclusion criteria were (a) children (2–12 y); (b) socially sustainable, family-, community-, or school-centered interventions targeting childhood obesity; (c) body weight, BMI (absolute or z-score), body fat, or waist circumference assessment; and (d) publication as a randomized controlled trial. Methodological quality was assessed using the 11-point PEDro scale. Results: Eleven studies were included (N = 39,902). Mean intervention duration was 27 months. BMI z-score and prevalence of overweight/obese children, assessed in 9 of 11 studies, were the most common anthropometric outcomes, followed by absolute BMI in 8 of 11 studies. Most studies were rated “fair” or “good” quality and indicated that school-, community- and family-oriented interventions may effectively improve anthropometric variables in pediatric obesity. Conclusions: Socially sustainable, multi-level interventions involving families, schools, and community systems appear to optimize anthropometric outcomes in childhood obesity by integrating health promotion into children’s daily social settings and reinforcing consistent, health-oriented norms and resources. Further studies employing independent, blinded evaluators, culturally sensitive components in interventions, and adequate participant adherence reports are required to enhance practical and clinical application.

## 1. Introduction

Childhood obesity has emerged as a critical global public health concern, recognized by the World Health Organization (WHO) as a growing epidemic [1]. According to WHO estimates, approximately 18%, or around 340 million, children aged ≥5 years were classified as overweight or obese in 2016 [1]. Definitions of overweight and obesity vary slightly across regions. The U.S. Centers for Disease Control and Prevention (CDC), for example, defines overweight as a body mass index (BMI) at or above the 85th percentile and below the 95th percentile for children of the same age and sex, while obesity is defined as a body mass index (BMI) at or above the 95th percentile [2]. In contrast, European guidelines typically classify individuals as overweight at or above the 85th percentile, and as obese at or above the 95th percentile, using similar age- and sex-specific BMI reference charts [3]. Regardless, the prevalence of childhood obesity is a major health concern, as excess body weight in early life often sets the stage for serious medical conditions traditionally associated with adulthood, including type 2 diabetes mellitus, hypertension, and hypercholesterolemia [4]. The physical health consequences of childhood obesity are extensive, encompassing cardiovascular complications, insulin resistance, metabolic syndrome, and a range of other conditions affecting the neurological, respiratory, musculoskeletal, hepatic, and renal systems [5]. Additionally, childhood obesity can significantly impact psychological well-being, contributing to low self-esteem, poor emotional and social functioning, and an overall reduced quality of life [6]. It is also linked to decreased academic performance [7] and significantly increases the risk of persistent obesity into adulthood [8], thereby posing serious health challenges in both developed and developing nations [4].

Obesity is a multifactorial condition influenced by a complex interplay of behavioral, environmental, socioeconomic, and genetic factors. Genetic predisposition contributes to obesity risk, particularly in cases where there is a family history of parental obesity [9]. Environmental key contributors include the increased intake of energy-dense, nutrient-poor foods; excessive consumption of sugar-sweetened beverages; and insufficient levels of physical activity [10]. Socioeconomic determinants, such as low parental education, urbanization, and the effects of modernization, also play a significant role in shaping dietary and lifestyle habits within families [10]. Moreover, sedentary behaviors, including prolonged screen time from television viewing and video gaming, have been strongly associated with the development of obesity in children [11]. In addition, the attitudes and perceptions of both children and their parents toward body weight, diet, and physical activity can significantly influence obesity-related behaviors and outcomes [11].

Given the high prevalence and multifactorial nature of childhood obesity, a wide range of interventions have been designed and implemented to address excess body weight within this population. Some of these initiatives focus on modifying external factors that influence health behaviors through social sustainability [12,13], such as strengthening social support networks, improving access to health education, and fostering inclusive environments where young individuals feel valued and respected. These strategies often target the primary settings that shape children’s daily lives—most notably, the family [14], community [15], and school environments [16]—recognizing their critical role in promoting healthier lifestyles and long-term behavioral change. Previous reviews on this matter [17,18,19,20] have primarily explored the overall effectiveness of behavioral, nutritional, and physical activity programs, or the environmental and policy determinants of obesity prevention. However, these reviews have not specifically addressed how social sustainability components, including family, school, and community engagement, cultural adaptability, and long-term social equity, may shape the effectiveness and durability of such interventions. Furthermore, while prior evidence syntheses encompassed a wide range of study designs, none focused exclusively on randomized controlled trials (RCTs) explicitly incorporating socially sustainable frameworks. Consequently, an evidence gap remains regarding the methodological rigor and anthropometric impact of RCT-based interventions that embed principles of social sustainability in pediatric obesity management. This review aims to fill that gap by systematically identifying and appraising randomized evidence that connects socially sustainable strategies to measurable anthropometric outcomes in children, thereby offering novel insights into how equity, inclusivity, and community integration can enhance the long-term effectiveness of obesity interventions.

The purpose of this scoping review was to examine the state of the science on interventions improving social sustainability and targeting anthropometric outcomes in overweight and obese children. It was hypothesized that such interventions would be effective in enhancing anthropometric parameters such as body weight, BMI, and fat mass in this population.

## 2. Materials and Methods

This review was performed following the methodology of the Preferred Reporting Items for Systematic reviews and Meta-Analyses extension for Scoping Reviews (PRISMA-ScR) statement [21] (Appendix A). The review protocol can be assessed at https://archive.org/details/osf-registrations-nrbua-v1 (accessed on 10 October 2025). The population, concept and context (PCC) framework according to Arksey and O’Malley [22] was also employed to support the research question formulation (Table 1), which led to the following research question: “*What is the existing evidence from RCTs on the effects of socially sustainable, family-, school-, or community-centered interventions on anthropometric outcomes in overweight and obese children?*”

### 2.1. Eligibility Criteria

Studies were considered eligible for analysis when the following inclusion criteria were met: (a) published as a full-text manuscript; (b) inclusion of children (2–12 years of age) participants (Population (P)); (c) socially sustainable, family-, community- or school-centered interventions targeting childhood obesity (Intervention (I)); (d) inclusion of body weight, BMI (absolute or z-score), body fat or waist circumference assessment (Outcome (O)); (e) published as a RCT (Study Design (S)); and (f) available in English, Portuguese, or Spanish. Socially sustainable interventions were considered strategies or programs designed to achieve their outcomes while promoting equity, inclusivity, and long-term social well-being and community acceptance, ensuring benefits are maintained without exacerbating social inequalities. Systematic reviews, meta-analyses, case reports, book chapters, abstracts, and conference proceedings were excluded. Studies that employed pharmacological or other non-community, school- or family-centered interventions aimed at treating or preventing childhood obesity were excluded. Studies not addressing anthropometric outcomes in overweight or obese children were also excluded. No limitations were set on the year of publication.

### 2.2. Search Strategy

A systematic electronic search was conducted following the PICOS (Population, Intervention, Control, Outcome, Study Design) principle. The literature search was carried out from 2 August to 1 September 2025 and included all studies published up to the latter date while encompassing the following databases: PubMed, Scopus, Cochrane Library, and Virtual Health Library. As a part of a secondary search, the reference list in each full text was scanned for additional studies. The search strategy was initiated using a wide range of keywords and combinations, and refined every single time, making it more specific to ensure that only the most relevant articles were retrieved during the screening process. Several Medical Subject Headings (MeSH) terms were used to facilitate searching. This method resulted in a search strategy consisting of multiple combinations of keywords related to population, intervention, and outcome across databases (Table 2). The keywords and MeSH terms were combined with Boolean operators (“AND/OR”). The search strategy was independently conducted by two authors (RP and VSP) to reduce selection bias. The reviewers were not blinded to any of the studies’ details. Disagreements between the authors were resolved through mutual consensus, and any inter-reviewer disagreements were settled by consensus with a third investigator (RCGC).

### 2.3. Study Coding and Data Extraction

All search results from databases were screened on title and abstract in Rayyan v.1.4.3, a computer software that facilitates and accelerates the work process in screening and selecting articles for inclusion in a systematic review. Studies were read and independently coded by two investigators (LVA and BHMB). The following data were extracted in an Excel template/spreadsheet: (a) author and year of publication; (b) descriptive information of participants’ characteristics, including the number of participants, age range, and nationality; (c) design (experimental groups/intervention and duration); (d) main results (compared to pre-intervention values). In the main results, primary outcomes included anthropometric data. Secondary outcomes were also extracted and included obesity-related behaviors, such as total amount of food intake, fruit ingestion, vegetable ingestion, fat ingestion, sugar-sweetened beverages ingestion, physical activity, and sedentary behavior. Whenever available, psychological variables were also extracted from the studies, such as quality of life (generally through the Pediatric Quality of Life Inventory, PedsQL), strengths and difficulties, and parental resource empowerment. In cases where studies lacked sufficient information regarding pre–post changes, the authors were contacted to provide the missing data. Whenever data were not acquired from authors, values were extracted from figures using WebPlotDigitizer v.5.2 online software, where applicable (https://apps.automeris.io/wpd/, accessed on 10 September 2025). Coding sheets were cross-checked between two reviewers (RP and VSP), with discussion and agreement over any observed differences, or if unresolved, adjudicated by a third independent reviewer (LVA).

### 2.4. Methodological Quality Assessment

For the assessment of methodological quality, the 11-point PEDro scale was used [23] and independently evaluated by two authors (RP and VSP). Discrepancies between reviewers were resolved by discussion, with a third independent reviewer consulted when consensus could not be reached (LVA). This scale is considered an internally valid and reliable tool for assessing the risk of bias for RCTs and clinical trials [19]. The PEDro scale consists of 11 items (yes/no questions). The first item assesses external validity and is not included in the total score. The items 2–11 assess internal validity; hence, the values of the PEDro scale range from 0 to 10. Results ranging from 9 to 10 were classified as “excellent”, 6 to 8 were classified as “good”, 4 to 5 were classified as “fair quality”, and 0 to 3 were classified as “poor quality”.

## 3. Results

### 3.1. Search Results

The search yielded 2241 (Pubmed, N = 1835; Cochrane Library, N = 316; Scopus, N = 24; Virtual Health Library, N = 66) possibly relevant references. From this number, 228 duplicates that appeared in one or more databases were excluded. After scrutinizing the abstracts and titles for relevance, we considered 44 full-texts appropriate for detailed review. A review of these studies revealed that only 11 studies [24,25,26,27,28,29,30,31,32,33,34] met all the inclusion criteria. Figure 1 presents a flow diagram of the search process. Ethics approval from the local institutional review board was noted in all the included studies.

### 3.2. Studies’ Description

The details of interventions from each study are presented in Table 3. The total number of participants across the studies was 39,902 participants, although one study alone was composed of 26,664 participants [26]. The total number of participants in control groups across the studies comprised 31,348 children, while experimental groups comprised 8554 participants. Studies’ duration lasted between 8 and 72 months (mean = 27 ± 20 months). Four studies employed exclusively community-centered programs [26,29,33,34]; three studies employed exclusively school-centered interventions [24,28,31]; two studies employed exclusively family-centered projects [30,32]; and two studies applied programs combining community-, school-, and family-orientation [25,27]. BMI z-score and percentage of overweight/obese children, evaluated in 8 out of 11 studies, were the most assessed anthropometric outcomes, followed by absolute BMI in 7 out of 11 studies. Seven studies reported a reduction in the percentage of overweight and/or obese children with socially sustainable interventions, while five reported a decrease in BMI z-score, three reported a decrease in absolute BMI, and one reported a reduction in fat mass percentage. In parallel, four studies did not detect any change in absolute BMI, while four and one studies reported, respectively, an absence of alteration in BMI z-score and percentage of overweight and/or obese children.

### 3.3. Methodological Quality

The PEDro score for the studies in this review ranged from 4 to 8 (mean = 5.5 ± 1.4), indicating “fair” to “good” methodological quality. Specifically, five studies were deemed to be of fair quality, and six studies were deemed to be of good quality. The details of quality analysis from each study are presented in Table 4. None of the articles received a score for Q5 and Q6, while only one received a score for Q7.

## 4. Discussion

A total of eleven RCTs were included in this review, comprising socially sustainable interventions targeting family, school, or community environments. Overall, the majority of trials reported beneficial anthropometric outcomes in pediatric obesity associated with socially sustainable interventions. The comparative efficacy between these approaches, however, remains inconclusive. Although the review did not definitively resolve which type of socially sustainable intervention model is superior, the collective evidence from the revised RCTs underscored meaningful insights into the efficacy of different components comprising socially sustainable interventions and their potential roles in the treatment of childhood obesity across diverse nationalities and settings.

### 4.1. School- and Community-Based Environmental Interventions

Several trials focused on environmental modifications at the school or community level. The pilot study by Warren and colleagues [24] evaluated a school-based intervention targeting 5- to 7-year-olds in Oxford, UK. Over approximately 14 months, children were randomly assigned to control, nutrition-only, physical activity–only, or combined arms. While all participants showed significant gains in nutrition knowledge, these did not translate into meaningful changes in overweight or obesity prevalence. Fruit and vegetable intake did increase modestly, but BMI remained unchanged, suggesting that improvements in knowledge and diet quality, while necessary, are insufficient by themselves to produce significant anthropometric changes.

In contrast, the “Romp & Chomp” intervention in Geelong, Australia (2004–2008), adopted a community-wide, multisetting approach focused on early childhood environments [26]. This multifaceted strategy, including community capacity building and environmental modifications in care and educational settings, yielded significant reductions in mean body weight, BMI, and BMI z-score among 3.5-year-olds, as well as a lower prevalence of overweight/obesity among both 2- and 3.5-year-old cohorts (reductions of 2.5 and 3.4 percentage points respectively, vs. ~0.7 percentage points in control groups). Behavioral changes reported by authors also included decreased consumption of packaged snacks and artificial fruit juice. These findings reflect the added potency of environmental strategies in achieving measurable outcomes in early childhood populations.

Likewise, the “LA Health” study [28] reported slight reductions in body fat among children exposed to school environmental modifications, though effect sizes were small and BMI z-score changes were not significant. Similarly, the “Fun ‘n Healthy in Moreland!” trial did not yield significant changes in BMI [31], though improvements in parameters other than anthropometric outcomes were demonstrated, such as in dietary behaviors, school policies, and child-reported wellbeing, as well as in the prevalence of overweight and obesity. On the other hand, a decrease in the prevalence of overweight and obesity was also observed in their control group [31]. In the same direction, the “Bright Start” trial in American Indian communities found no mean BMI differences, despite a 10% reduction in the prevalence of overweight, parallel to a significant increase in overweight prevalence in their control group, suggesting benefits in incidence prevention [27].

On the other hand, “Shape Up Somerville” showed significant reductions in BMI z-scores during its first year, attributed to broad community engagement and policy/environmental changes [25]. Likewise, the “TX CORD” trial reported that community-based programs with higher contact hours achieved greater short-term improvements in BMI and fat mass among underserved populations compared to primary care interventions [29], though sustaining effects remained a challenge. Finally, the “Children’s Healthy Living (CHL) Program”, across Pacific jurisdictions, achieved meaningful reductions in absolute BMI, BMI z-score, and overweight/obesity prevalence immediately post-intervention [33], while a sustained benefit was shown in overweight and obesity prevalence over six years [34]. These findings suggest that multilevel, culturally contextualized, and long-duration interventions are more likely to achieve measurable population-level effects than shorter-term, school-only approaches.

### 4.2. Family-Based Interventions

More intensive clinical and family-centered strategies demonstrated consistent efficacy. The “Multi-Site Family-Based Treatment (FBT)” trial showed that enhanced social facilitation maintenance programs improved the prevalence of overweight children, but not BMI z-score relative to education controls, with treatment dose influencing efficacy [32]. Conversely, the “Connect for Health” trial demonstrated modest but significant BMI z-score improvements with enhanced primary care interventions [30], though the addition of personalized health coaching to families did not significantly augment effects.

### 4.3. General Overview

A central observation across the revised studies was the limited effect on BMI when interventions relied solely on educational or environmental changes of modest intensity and duration. Stronger effects appeared when programs integrated multiple components—school, family, and broader community systems—especially when maintained over time. Additionally, sustainability emerges as a critical factor: while many interventions showed short-term benefits, only the CHL Program demonstrated durable effects beyond the intervention period. Another recurring finding is that behavioral and environmental improvements (e.g., healthier diet, increased activity opportunities, policy changes) were often achieved even when anthropometric outcomes remained unchanged. These intermediate outcomes are meaningful in shaping supportive contexts for long-term obesity prevention and treatment, particularly in high-risk and underserved populations.

### 4.4. Limitations

Despite the broad diversity of populations studied across the trials (e.g., U.S. children, American Indian communities, Pacific Islanders), some of the studies were limited by modest sample sizes and relatively short intervention or follow-up periods. Heterogeneity in outcome measures and intervention intensity also impairs direct comparisons. In addition, one large-scale trial [26] alone accounted for more than two-thirds of all participants, which may have disproportionately influenced the overall synthesis and warrants caution when interpreting the findings.

A significant limitation identified across many of the studies included was the lack of consideration for cultural factors among participants, and as shown in Table 3, the studies were conducted across several distinct locations. Variables such as ethnicity, religious beliefs and practices, and neighborhood characteristics are well-documented to influence health behaviors and lifestyle choices in children and their families, factors that are closely linked to the prevalence of childhood obesity [35]. For example, a study by Chen et al. [36] demonstrated that culturally tailored, web-based behavioral interventions targeting Chinese American adolescents were more effective in promoting healthy lifestyle changes and reducing obesity risk. This finding underscores the importance of incorporating culturally sensitive components into intervention design to enhance both relevance and efficacy across diverse populations.

Furthermore, lack of adherence reports also emerged as a common limitation across all the studies. To enhance participant engagement, some researchers reported using different strategies, including email reminders, telephone follow-ups, and financial incentives. Nevertheless, one cannot assume that the adherence rate was necessarily high, especially since most of the reviewed studies included very long durations. In this sense, Novotny et al. [33] demonstrated in their first 24-month study that a multijurisdictional, multilevel, multicomponent community-based program was effective in improving anthropometric outcomes such as absolute BMI, BMI z-score, waist circumference, and the percentage of overweight and obese participants. On the other hand, during their 4-year follow-up [34], despite a sustained benefit still observed in overweight prevalence, no improvements were identified in absolute BMI and BMI z-score compared to the measurement taken after 24 months of intervention. In fact, a statistically significant increase in waist circumference was observed in the intervention group, suggesting limited long-term efficacy. These findings suggest that strategies need to be developed to maximize sustained engagement, thus optimizing and maintaining treatment efficacy.

Lastly, as shown in Table 4, none of the studies met items Q5 and Q6 of the PEDro scale, and only one met Q7. The absence of compliance with Q5 and Q6 is understandable in studies involving behavioral interventions (e.g., exercise training, specific diets), as blinding both the participants and the intervention providers is often impractical. However, the lack of compliance with Q7 is less justifiable, since it would have been feasible to assign outcome assessors blinded to the intervention type or experimental groups when evaluating anthropometric or other parameters. The absence of blinded assessors likely increased the risk of bias in most of the reviewed studies, underscoring the importance of employing blinded evaluators in future research to enhance methodological rigor. Moreover, although some of the included studies [24,26,27] did not explicitly state their eligibility criteria, all clearly targeted preschool and school-aged children and employed comparable intervention settings and outcome measures. Therefore, their inclusion does not compromise the generalizability of the findings. On the contrary, retaining these studies strengthens the representativeness of the evidence base by encompassing a broader range of socially sustainable, real-world interventions consistent with the review’s scope.

## 5. Conclusions

Across diverse pediatric populations and settings, childhood obesity management requires comprehensive, multi-component strategies. School- and family-based programs achieved notable improvements in dietary behaviors and physical activity opportunities; however, their effects on anthropometric outcomes were generally small or inconsistent. More intensive, multilevel community-based interventions produced greater short-term improvements in anthropometric parameters, yet sustaining these benefits over time remains challenging. These results underscore that scalable and equitable solutions to childhood obesity must integrate efforts across schools, families, healthcare systems, and communities, with long-term investment to support sustainability and health promotion.

Future research should aim to strengthen adherence, reduce socioeconomic disparities, and extend intervention benefits beyond short-term gains toward lasting improvements in child health. To enhance practical and clinical application, studies should incorporate (a) blinded research personnel to reduce assessment bias; (b) culturally sensitive components to ensure relevance and engagement; and (c) adequate reporting of participant adherence to inform fidelity and reproducibility.

## Figures and Tables

**Figure 1 healthcare-13-02932-f001:**
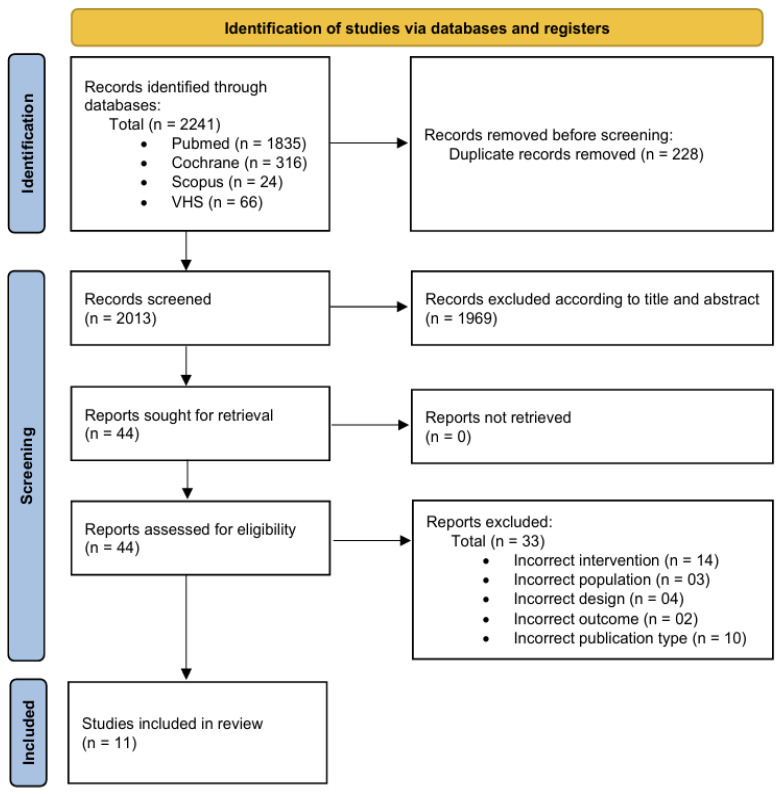
Flow diagram of the search process.

**Table 1 healthcare-13-02932-t001:** The population, concept, and context (PCC) framework used to guide the research question and inclusion criteria.

Element	Description	Application in the Review
Population (P)	Children aged 2–12 years who are overweight or obese	The review included studies with pediatric participants (2–12 years old) classified as overweight or obese according to age- and sex-specific BMI criteria.
Concept (C)	Socially sustainable interventions designed to prevent or treat childhood obesity.	Interventions that incorporate elements promoting equity, inclusivity, community engagement, and long-term social well-being, such as family-, school-, or community-based programs aimed at fostering sustainable health-related behaviors.
Context (C)	Family, school, and community environments where health promotion and obesity prevention occur.	Studies were included if they evaluated interventions implemented in real-world social settings (home, schools, or community programs) and assessed anthropometric outcomes within those contexts.

**Table 2 healthcare-13-02932-t002:** Search strategy for the PubMed database.

Keywords				
P (Population)		Intervention		Outcome
“child nutrition disorders” [Mesh] OR “pediatric obesity” [Mesh]	AND	“sustainable development” [Mesh] OR “sustainable development indicators” OR “social environment” [Mesh] OR “social sustainability”	AND	“body weight” [Mesh] OR “body mass index” [Mesh] OR “waist circumference” [Mesh] OR “anthropometric outcome”

Applied filters: ‘Abstract’ and ‘full text’ (text availability); ‘randomized controlled trial’ (article type); ‘English’, ‘Portuguese’ and ‘Spanish’ (article language); ‘humans’ (species); ‘female’ and ‘male’ (sex); ‘child’ (age).

**Table 3 healthcare-13-02932-t003:** Summary of the basic details from the included studies.

Authors	Participants’ Characteristics	Design	Results
**Butte et al. [29]**	549 North American obese (BMI ≥ 85th percentile) children from 12 primary care clinics stratified in 3 age groups (2–5 y; 6–8 y; 9–12 y)	12-month (Intensive Phase = 3 months; Transition Phase = 9 months) community-centered weight management program (Intervention group; n = 315) vs. primary care-centered program (Control group; n = 234)	Intervention group (including 2–5 y, 6–8 y and 9–12 y), Intensive Phase: ↓BMI (*p* < 0.05), ↓%fat mass (*p* < 0.05), ↑PedsQL (*p* < 0.05), ↓SDQ (*p* < 0.05) Intervention group (including 2–5 y, 6–8 y and 9–12 y), Transition Phase: ↔BMI (*p* > 0.05), ↑%fat mass (*p* < 0.05), ↔PedsQL (*p* > 0.05), ↔SDQ (*p* > 0.05) Control group (including 2–5 y, 6–8 y and 9–12 y), Intensive Phase: ↔BMI (*p* > 0.35), ↔%fat mass (*p* > 0.05), ↑PedsQL (*p* < 0.05), ↓SDQ (*p* < 0.05) Control group (including 2–5 y, 6–8 y and 9–12 y), Transition Phase: ↑BMI (*p* < 0.05), ↑%fat mass (*p* < 0.05), ↑PedsQL (*p* < 0.05), ↔SDQ (*p* > 0.05)
**de Silva-Sanigorski et al. [26]**	26,664 Australian children from local government areas in Victoria (0–5 y)	4-year community-wide, multisetting, multistrategy program in CoGG and BoQ (0–2 y, n = 1457; 3.5–5 y, n = 1082) vs. LGA across the rest of Victoria (0–2 y, n = 14,924; 3–5 y, n = 9201)	Intervention group: ↓weight (*p* < 0.05), ↓BMI (*p* < 0.05), ↓BMI z-score (*p* < 0.05), ↓%overweight/obese children (*p* < 0.05) (both 0–2 y and 3.5–5 y), ↑fruit ingestion (*p* = 0.03), ↑vegetable ingestion (*p* < 0.001), ↓sugar-sweetened beverages (*p* = 0.005) Control group: ↔weight (*p* > 0.05), ↔BMI (*p* > 0.05), ↔BMI z-score (*p* > 0.05), ↔%overweight/obese children (*p* > 0.05) (both 0–2 y and 3.5–5 y), ↑fruit ingestion (*p* = 0.03), ↑vegetable ingestion (*p* < 0.001), ↓sugar-sweetened beverages (*p* = 0.005)
**Economos et al. [25]**	1178 North American Grade 1, 2, and 3 public school children from 3 communities in Somerville (6–9 y)	8-month community-, family and school-centered weight gain prevention program (Intervention group; n = 385) vs. Control communities (Control group; n = 793)	Intervention group: ↓BMI z-score (*p* = 0.001) Control group: ↔BMI z-score (*p* > 0.05)
**Novotny et al. [33]**	4042 US-Affiliated Pacific region overweight (BMI ≥ 85th–94th percentile), obese (BMI ≥ 95th percentile), and acanthosis nigricans children (2–8 y)	24-month multijurisdictional, multilevel, multicomponent community-based program (n = 1342) vs. control group (n = 1295) vs. Temporal group (n = 1405)	Intervention group: ↓BMI z-score (*p* = 0.006), ↓waist circumference (*p* = 0.02), ↓%overweight/obese children (*p* = 0.02), ↔fruit ingestion (*p* = 0.07), ↔vegetable ingestion (*p* = 0.73), ↔sugar-sweetened beverages (*p* = 0.09) Control group: ↔BMI z-score (*p* = 0.84), ↑waist circumference (*p* < 0.001), ↔%overweight/obese children (*p* = 0.35), ↔fruit ingestion (*p* = 0.09), ↔vegetable ingestion (*p* = 0.89), ↔sugar-sweetened beverages (*p* = 0.40) Temporal group: ↔BMI z-score (*p* = 0.42), ↔waist circumference (*p* = 0.06), ↔%overweight/obese children (*p* = 0.84), fruit ingestion (not measured), vegetable ingestion (not measured), sugar-sweetened beverages (not measured)
**Novotny et al. [34]**	1469 US-Affiliated Pacific region overweight (BMI ≥ 85th–94th percentile), obese (BMI ≥ 95th percentile), and acanthosis nigricans children (2–8 y)	6-year multijurisdictional, multilevel, multicomponent community-based program (n = 500) vs. control group (n = 479) vs. Temporal group (n = 490)	Intervention group: ↓BMI z-score (*p* = 0.02), ↔waist circumference (*p* = 0.06), ↓%overweight/obese children (*p* = 0.05), ↔fruit ingestion (*p* = 0.71), ↔vegetable ingestion (*p* = 0.99), ↔sugar-sweetened beverages (*p* = 0.51) Control group: ↑BMI z-score (*p* = 0.05), ↑waist circumference (*p* < 0.0001), ↑%overweight/obese children (*p* = 0.009), ↔fruit ingestion (*p* = 0.08), ↔vegetable ingestion (*p* = 0.13), ↔sugar-sweetened beverages (*p* = 0.10) Temporal group: ↔BMI z-score (*p* = 0.16), ↑waist circumference (*p* = 0.0005), ↔%overweight/obese children, fruit ingestion (not measured), vegetable ingestion (not measured), sugar-sweetened beverages (not measured).
**Story et al. [27]**	454 North American kindergarten children (5–6 y) from 14 schools on the Pine Ridge Reservation in South Dakota	14-week family- and school-centered weight gain prevention program (Intervention group; n = 267) vs. Control group (n = 187)	Intervention group: ↔BMI (*p* = 0.057), ↔BMI z-score (*p* = 0.90), ↔%fat mass (*p* = 0.12), ↓%overweight children (*p* = 0.01), ↔obese children (*p* = 0.50), ↓sugar-sweetened beverages (*p* = 0.02), ↓fat ingestion (*p* = 0.004). Control group: ↔BMI (*p* > 0.05), ↔BMI z-score (*p* > 0.05), ↔%fat mass (*p* > 0.05), ↑%overweight children (*p* > 0.05), ↔obese children (*p* > 0.05), ↔sugar-sweetened beverages (*p* > 0.05), ↔fat ingestion (*p* > 0.05).
**Taveras et al. [30]**	664 North American overweight and obese (BMI ≥ 85th percentile) children (2–12.9 y)	1-year enhanced primary care plus family- and contextually centered individual health coaching (Intervention group; n = 336) vs. enhanced primary care (Control group; n = 328)	Intervention group: ↓BMI z-score (*p* < 0.05), ↓%overweight children (*p* < 0.05), ↓%obese children (*p* < 0.05), ↑PedsQL (*p* < 0.05), ↑Parenteral resource empowerment (*p* < 0.05). Control group: ↓BMI z-score (*p* < 0.05), ↓%overweight children (*p* < 0.05), ↓%obese children (*p* < 0.05), ↔PedsQL (*p* > 0.05), ↔Parenteral resource empowerment (*p* > 0.05).
**Warren et al. [24]**	218 British children (5–7 y) from 3 primary schools in Oxford	20-week school-centered weight gain prevention program through Nutrition (Intervention group 1; n = 56) vs. Physical activity (Intervention group 2; n = 54) vs. Physical activity plus Nutrition (Intervention group 3; n = 54) vs. Education (Control group; n = 54)	Intervention group 1: ↔BMI (*p* > 0.05), ↔%overweight children (*p* > 0.05), ↔obese children (*p* > 0.05), ↑fruit ingestion (*p* < 0.05), ↔physical activity (*p* > 0.05). Intervention group 2: ↔BMI (*p* > 0.05), ↔%overweight children (*p* > 0.05), ↔obese children (*p* > 0.05), ↔fruit ingestion (*p* > 0.05), ↑physical activity (*p* < 0.05). Intervention group 3: ↔BMI (*p* > 0.05), ↔%overweight children (*p* > 0.05), ↔obese children (*p* > 0.05), ↔fruit ingestion (*p* > 0.05), ↑physical activity (*p* < 0.05). Control group: ↔BMI (*p* > 0.05), ↔%overweight children (*p* > 0.05), ↔obese children (*p* > 0.05), ↑fruit ingestion (*p* < 0.05), ↔physical activity (*p* > 0.05)
**Waters et al. [31]**	2806 Australian children (5–12 y) from 22 primary schools in Moreland	3.5-year school-centered weight gain prevention program (Intervention group; n = 1346) vs. normal school programs (Control group; n = 1460)	Intervention group: ↔BMI (*p* > 0.05), ↓BMI z-score (*p* < 0.05), ↔waist circumference (*p* > 0.05), ↓%overweight children (*p* < 0.05), ↓obese children (*p* < 0.05), ↑fruit ingestion (*p* < 0.05), ↔physical activity (*p* > 0.05) Control group: ↔BMI (*p* > 0.05), ↓BMI z-score (*p* < 0.05), ↔waist circumference (*p* > 0.05), ↓%overweight children (*p* < 0.05), ↓obese children (*p* < 0.05), ↔fruit ingestion (*p* > 0.05), ↔physical activity (*p* > 0.05)
**Wilfley et al. [32]**	160 North American overweight or obese (BMI ≥ 85th percentile) children (7–11 y)	12-month family-based behavioral weight loss program of low (Intervention group 1; n = 54) and high dose (Intervention group 2; n = 55) vs. weight-control education program (Control group; n = 51)	Intervention group 1: ↔BMI z-score (*p* = 0.24), ↓%overweight children (*p* < 0.001) Intervention group 2: ↔BMI z-score (*p* = 0.81), ↓%overweight children (*p* = 0.02) Control group: ↔BMI z-score (*p* > 0.05), ↔%overweight children (*p* > 0.05)
**Williamson et al. [28]**	1697 North American Grades 4–6 children from 17 school systems in Louisiana (9–11 y)	28-month school-centered healthy habits program (Intervention group 1; n = 612) vs. school-centered healthy habits program + internet-based educational approach (Intervention group 2; n = 638) vs. Control group (n = 447)	Intervention group 1: ↔BMI z-score (*p* > 0.05), ↔%fat mass (*p* > 0.05), ↔food intake (*p* > 0.05), ↔physical activity (*p* > 0.05), ↔sedentary behaviour (*p* > 0.05) Intervention group 2: ↔BMI z-score (*p* > 0.05), ↔%fat mass (*p* > 0.05), ↔food intake (*p* > 0.05), ↔physical activity (*p* > 0.05), ↔sedentary behaviour (*p* > 0.05) Control group: ↔BMI z-score (*p* > 0.05), ↔%fat mass (*p* > 0.05), ↔food intake (*p* > 0.05), ↔physical activity (*p* > 0.05), ↔sedentary behaviour (*p* > 0.05)

**Legend:** BMI = body mass index, PedsQL = Parent-reported quality of life, SDQ = Strengths and Difficulties Questionnaire. The symbol ↑ indicates a significant increase; the symbol ↓ indicates a significant decrease; the symbol ↔ indicates a lack of significant difference.

**Table 4 healthcare-13-02932-t004:** Studies’ quality through the 11-point Physiotherapy Evidence Database (PEDro) scale.

Reference	Q1	Q2	Q3	Q4	Q5	Q6	Q7	Q8	Q9	Q10	Q11	Total
Butte et al. [29]	Yes	✔	✔	✔	X	X	X	X	✔	✔	✔	6
De Silva-Sanigorski et al. [26]	No	✔	X	X	X	X	X	X	✔	✔	✔	4
Economos et al. [25]	Yes	✔	X	✔	X	X	X	X	✔	✔	✔	5
Novotny et al. [33]	Yes	✔	X	✔	X	X	X	✔	✔	✔	✔	6
Novotny et al. [34]	Yes	✔	X	✔	X	X	X	X	✔	✔	✔	5
Story et al. [27]	No	✔	X	✔	X	X	X	X	✔	✔	✔	5
Taveras et al. [30]	Yes	✔	✔	✔	X	X	X	✔	✔	✔	✔	7
Warren et al. [24]	No	✔	X	✔	X	X	X	X	✔	✔	✔	5
Waters et al. [31]	Yes	✔	✔	✔	X	X	✔	✔	✔	✔	✔	8
Wilfley et al. [32]	Yes	✔	✔	✔	X	X	X	✔	✔	✔	✔	7
Williamson et [28]	Yes	✔	X	✔	X	X	X	X	✔	✔	✔	5

**Legend:** The symbol ✔ means meeting the question/criterion, while the symbol X means not meeting the question/criterion. PEDro Scale: Q1. Eligibility criteria stated; Q2. Random allocation; Q3. Concealed allocation; Q4. Groups similar at baseline; Q5. Participants blinded; Q6. Therapists blinded; Q7. Assessors blinded; Q8. ≥85% follow-up; Q9. Intention-to-treat analysis; Q10. Between-group comparisons reported; Q11. Point estimates and variability provided.

## Data Availability

Data sharing not applicable as no datasets were generated and/or analysed for this study.

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
