# Peer review of "Socially Sustainable Interventions for Childhood Obesity Management: A Scoping Review of Randomized Controlled Trials"

_healthcare, 2025, doi:10.3390/healthcare13222932_

Round 1
Reviewer 1 Report
Comments and Suggestions for Authors
General comments:
Line 23-24: It is important to explicitly display the date of the article search.
Line 25: “human participants”, delete. “socially sustainable” in the same line should be clearly defined.
Line 30-31: “mostly comprising children” and "mean studies’ duration was 25±20 months"- meaning unclear.
Line 35-37: The conclusion could be improved for clarity. It is unclear how anthropometric outcomes in childhood obesity may be enhanced from socially sustainable family-, school- and community-oriented interventions.
Line 38-39: These terms "blinded research personnel" and "adequate adherence report" are vague should be described in details.
Refs# 2,3 & 19 are old-please update.
Line 88-92: A clear description of the evidence gap that this review is filling is required. No clear rationale on why this review is of importance in light of previous scoping reviews (Obes Rev. 2022 May;23(5):e13427; Int J Pediatr. 2025 Apr 28;2025:8871022; Obes Rev. 2021 Nov 28;23(3):e13378; Obes Sci Pract. 2021 Dec 10;8(3):371–386).
"Socially sustainable interventions" must be clearly defined at the beginning of methods.
The PCC framework (population, concept, and context) is recommended in a Table as a guide to identify the main concepts in the research question.
Line 111 & 115-116: The date of the article search should be displayed clearly. It is unclear whether the period from August to September 2025 was only searched.
Line 152-158: Clarify how discrepancies were resolved. Clarify the role of other authors.
Figure 1: Please used another term or expression instead of “wrong”.
Line 178: “Individuals”???
Table 1: Data extraction requires a lot of planning. Please include additional columns. Significant P-value should be also presented in the results.
In conclusions, make the information more structured by keeping the recommendations and findings apart. Consider carefully the findings' ramifications and how they might affect subsequent interventions.
Comments on the Quality of English LanguagePlease avoid using the term “we” throughout the paper (e.g., we used, we contacted…etc.).
Author Response
Dear Reviewer,
Thank you for your comments.
Our best performance was done to change the manuscript accordingly to to your comments; or otherwise your comments were answered.
The English Language was also reviewed. You will find the response to each comment in the attached document. We hope to have met the standard required for publication, although we are happy to perform any further amendments in case of necessity.
Kind regards,
The authors.

Reviewer 2 Report
Comments and Suggestions for Authors
I appreciate the authors for the manuscript entitled “Socially sustainable interventions to prevent or treat childhood obesity: a scoping review of randomized controlled trials,” which provides insights into social interventions for the management of childhood obesity. The manuscript is well written and scientifically sound. However, I made numerous suggestions to improve the quality of the manuscript.
Title
It’s challenging and complex to prevent obesity, and I suggest the author replace “Prevent or treat” with the term “managing” in the title.
Introduction
The authors clearly addressed the multifactorial influence of obesity.
Method
In the search strategy, the term "anthropometric outcome" is missing. In some articles, the authors mention anthropometric outcomes instead of body weight, BMI, and waist circumference. The authors need to clarify it.
Line 152: Most readers are not familiar with the PEDro scale. So, I suggest that the authors describe the eleven items included in the PEDro scale.
Results
Lines 162-163 and Figure 1: I noticed a mistake in the summing of PubMed, N=1800; Cochrane Library, N=316; Scopus, N=24; and Virtual Health Library, N=66, which equals 2182. But the actual total is 2206. I suggest that the authors modify these sentences and Figure 1. Please check carefully if any studies are missing since the authors mentioned only 2182 instead of the actual 2206.
Table 1: Warren et al. 2003 [19]. There is a mistake in summing the total number of participants. The authors mentioned the “213 British children..”; however, the summary of Intervention group 1 (n=56) vs. Physical activity (Intervention group 2; n=54) vs. Physical activity plus Nutrition (Intervention group 3; n=54) vs. Education (Control group; n=54) actually yields 218. Please correct it.
Table 1: Novotny et al. 2018 [29]. The authors mentioned that the “4048 US-Affiliated Pacific..”. There is another mistake in the total number of participants. community-based program (n=1342) vs. control group (n=1295) vs. Temporal group (n=1405) actually yields 4042. Please correct it.
Table 1: Please describe the symbols ↓, ↑, ↔ under the footnotes in Table 1.
Table 2: Please describe Q1-Q11 under the footnotes of Table 2.
Methodological quality: None of the articles received a score for Q5 and Q6. Please highlight this and discuss it in the discussion section.
Discussion
Line 207: “A total of twelve randomized controlled trials..” One study was found to have poor quality [De Silva-Sanigorski et al. (2010)]; still, the authors consider it in this review. Please clarify it in the discussion.
Also, three studies have no external validity [De Silva-Sanigorski et al., 2010; Story et al., 2012; Warren et al., 2003]. Please clarify in the discussion whether it affects generalizability and whether these are worth using in this review.
Author Response
Dear Reviewer,
Thank you for your comments.
Our best performance was done to change the manuscript accordingly to to your comments; or otherwise your comments were answered.
You will find the response to each comment in the attached document.
We hope to have met the standard required for publication, although we are happy to perform any further amendments in case of necessity.
Best regards,
The authors.

Round 2
Reviewer 1 Report
Comments and Suggestions for Authors
Line 23 & Line 141-142: This needs to be clarified. Please include a period of time encompassed by the search (starting year chose to be included in this review). For example, from Jan 2001 to Sep 2025.
Author Response
Comment 1: Line 23 & Line 141-142: This needs to be clarified. Please include a period of time encompassed by the search (starting year chose to be included in this review). For example, from Jan 2001 to Sep 2025.
Response 1: This has now been amended.
Reviewer 2 Report
Comments and Suggestions for Authors
The authors revised the manuscript according to my recommendations, and I believe it has improved in quality.
I have no further comments.
Best regards
Author Response
Comment 1: The authors revised the manuscript according to my recommendations, and I believe it has improved in quality. I have no further comments. Best regards.
Response 1: Thank you very much for all your valuable comments and criticisms.